# Direct factor Xa inhibitors and the risk of cancer and cancer mortality: A Danish population-based cohort study

**Floris Bosch**[1,2,3]*, **Erzsébet Horváth-Puhó**[4], **Suzanne C. Cannegieter**[5,6], **Nick van Es**[2,3], **Henrik T. Sørensen**[4]

**1** Department of Internal Medicine, Tergooi Hospitals, Hilversum, the Netherlands, **2** Amsterdam UMC location University of Amsterdam, Department of Vascular Medicine, Amsterdam, the Netherlands, **3** Amsterdam Cardiovascular Sciences, Pulmonary Hypertension & Thrombosis, Amsterdam, the Netherlands, **4** Department of Clinical Epidemiology, Aarhus University and Aarhus University Hospital, Aarhus, Denmark, **5** Department of Clinical Epidemiology, Leiden University Medical Center, Leiden, the Netherlands, **6** Department of Thrombosis and Hemostasis, Leiden University Medical Center, Leiden, the Netherlands

* f.t.bosch@amsterdamumc.nl

**Data Availability Statement:** All relevant data are within the manuscript and its Supporting Information files.

## Abstract

### Background

Preclinical animal studies have suggested that myeloid cell–synthesized coagulation factor X dampens antitumor immunity and that rivaroxaban, a direct factor Xa inhibitor, can be used to promote tumor immunity. This study was aimed at assessing whether patients with atrial fibrillation taking direct factor Xa inhibitors have lower risk of cancer and cancer-related mortality than patients taking the direct thrombin inhibitor dabigatran.

### Methods and findings

This nationwide population-based cohort study in Denmark included adult patients with atrial fibrillation and without a history of cancer, who started taking a factor Xa inhibitor or dabigatran between 2011 and 2015. Data on medical history, outcomes, and drug use were acquired through Danish healthcare registries. The primary outcome was any cancer. Secondary outcomes were cancer-related mortality and all-cause mortality. Outcome events were assessed during 5 years of follow-up in an intention-to-treat analysis. The propensity score-based inverse probability of treatment weighting was used to compute cumulative incidence and subdistribution hazard ratios (SHRs) and corresponding 95% confidence intervals (CIs), with death as a competing event. Propensity scores were estimated using logistic regression and including in the model sex, age group at index date, comorbidities, and use of comedications. A total of 11,742 patients with atrial fibrillation starting a factor Xa inhibitor and 11,970 patients starting dabigatran were included. Mean age was 75.2 years (standard deviation [SD] 11.2) in the factor Xa cohort and 71.7 years (SD 11.1) in the dabigatran cohort. On the basis of the propensity score-weighted models, after 5 years of follow-up, no substantial difference in the cumulative incidence of cancer was observed between the factor Xa inhibitor (2,157/23,711; 9.11%, 95% CI [8.61%,9.63%]) and dabigatran (2,294/

**Funding:** The author(s) received no specific funding for this work.

**Competing interests:** The Department of Clinical Epidemiology receives funding for other studies from companies in the form of research grants to (and administered by) Aarhus University (none of those studies have any relation to the present study). N.v.E. reports receiving consultancy fees from Bayer, LEO Pharma, and Daiichi Sankyo, which were transferred to his institution (none of those fees have any relation to the present study). The remaining authors declare no competing financial interests.

**Abbreviations:** ATC, Anatomical Therapeutic Chemical; CI, confidence interval; COVID-19, Coronavirus Disease 2019; DNPR, Danish National Patient Registry; DOAC, direct oral anticoagulant; IPTW, inverse probability of treatment weighted; IQR, interquartile range; LMWH, low-molecular-weight heparin; NSAID, nonsteroidal anti-inflammatory drug; PAR2, protease activated receptor 2; RCT, randomized controlled trial; SD, standard deviation; SHR, subdistribution hazard ratio; TGF-β1, transforming growth factor-β1.

23,715; 9.68%, 95% CI [9.14%,10.25%]) groups (SHR 0.94, 95% CI [0.89,1.00], *P* value 0.0357). We observed no difference in cancer-related mortality (factor Xa inhibitors cohort 1,028/23,711; 4.33%, 95% CI [4.02%,4.68%]. Dabigatran cohort 1,001/23,715; 4.22%, 95% CI [3.83%,4.66%]; SHR 1.03, 95% CI [0.94,1.12]), but all-cause mortality was higher in the factor Xa inhibitor cohort (factor Xa inhibitors cohort 7,416/23,711; 31.31%, 95% CI [30.37%,32.29%]. Dabigatran cohort 6,531/23,715; 27.56%, 95% CI [26.69%,28.45%]; HR 1.17, 95% CI [1.13,1.21]). The main limitations of the study were the possibility of residual confounding and the short follow-up period.

## Conclusions

In this population based cohort study, factor Xa inhibitor use was not associated with an overall lower incidence of cancer or cancer-related mortality when compared to dabigatran. We did observe an increase in all-cause mortality in the factor Xa inhibitor cohort.

## Author summary

### Why was this study done?

- A preclinical study in mice with breast cancer and fibrosarcoma showed that factor X dampens antitumor immunity and that factor Xa inhibitor promote tumor immunity.

- Whether factor Xa inhibition is associated with decreased cancer incidence and cancer-related mortality in humans is unknown.

### What did the researchers do and find?

- We assessed cancer incidence during 5 years of follow-up in patients with atrial fibrillation in Denmark using a factor Xa inhibitor (*n* = 11,742) or a thrombin inhibitor (dabigatran) (*n* = 11,970) for stroke prevention.

- No substantial difference in the cumulative incidence of cancer was observed between the factor Xa inhibitor (9.11%, 95% CI [8.61%,9.63%]) and dabigatran (9.68%, 95% CI [9.14%,10.25%]) groups (SHR 0.94, 95% CI [0.89,1.00].

- No difference in cancer-related mortality (SHR 1.03, 95% CI [0.94,1.12]) was observed, but all-cause mortality was higher in the factor Xa inhibitor cohort (HR 1.17, 95% CI [1.13,1.21]).

### What do these findings mean?

- Factor Xa inhibitor use for atrial fibrillation did not appear to significantly reduce cancer risk compared to dabigatran use.

- The main limitations of the study were the possibility of residual confounding and the short follow-up period.

## Introduction

Cancer activates the hemostatic system, thereby promoting tumor growth and metastasis [1]. Components of the coagulation cascade have been suggested to play important roles in tumor spread and cancer progression [2,3]. For example, tissue factor promotes tumor progression in breast cancer, fibrosarcoma, colon carcinoma, melanoma, and pancreatic cancer [4–6], possibly by binding factor VIIa and subsequently cleaving protease activated receptor 2 (PAR2), thereby inducing the release of proangiogenic factors, and promoting cell migration and metastasis [5]. Whether anticoagulants confer a benefit in patients with cancer by slowing tumor growth and metastasis has long been debated and different randomized controlled trials on low-molecular-weight heparin (LMWH) in patients with cancer showed both a beneficial as well as no effect on survival [7–11]. Finally, a systematic review and meta-analysis on 9 randomized controlled trials (RCTs) showed no effect on survival [12].

A recent study on breast cancer and fibrosarcoma in mice has indicated that monocytes and macrophages produce factor X in the tumor microenvironment, where it dampens antitumor immunity by signaling through PAR2 [13]. Specifically, factor X promotes tumor immune evasion by recruiting immune-suppressive neutrophils and regulatory T cells. Pharmacological blockade of this factor Xa-PAR2 axis by the factor Xa inhibitor rivaroxaban has been found to decrease the risk of cancer progression in breast cancer and fibrosarcoma by promoting antitumor immunity, with efficacy comparable to that of immune checkpoint inhibition [13].

We performed a nationwide cohort study of patients with newly diagnosed atrial fibrillation initiating treatment with direct oral anticoagulants (DOACs) for stroke prevention. Using an active comparator, new user design, we compared patients prescribed factor Xa inhibitors (rivaroxaban, apixaban, or edoxaban) with those receiving the direct thrombin inhibitor dabigatran. Given the factor X–specific cancer effects, we hypothesized that incident cancer, cancer progression, and death from cancer would be lower in patients taking direct factor Xa inhibitors rather than dabigatran.

## Methods

### Setting, design, and data sources

This study was executed according to a prospective protocol, which can be found in the Supporting information (S1 Appendix). In Denmark, all residents have free access to a universal tax-supported healthcare system [14]. Healthcare data from Danish residents are collected in national medical and administrative registries. Data on demographics, medical history, medication use, and death are recorded in different registries, and their records are linked through the unique civil register numbers that are given to each resident in Denmark at the time of birth or immigration, and recorded in the Danish Civil Registration System.

This study used several Danish registries as data sources. The Danish National Patient Registry (DNPR) has collected data on all hospitalizations in Denmark since 1977 [15] and on outpatient clinic and emergency department visits since 1995. Diagnoses have been coded via the 10th revision of the *International Statistical Classification of Diseases and Related Health Problems* (ICD-10) system since January 1994 and were coded according to the ICD-8 classification system before January 1994 [15]. The Danish National Prescription Registry records data on dispensed medications and dates of prescription from 1995 onward, by using the Anatomical Therapeutic Chemical (ATC) coding system [16]. The Danish Cancer Registry contains records of all incidences of malignant neoplasms in the Danish population, according to the ICD-

10 coding system, from 1978 onward [17]. Causes of death are recorded in the Danish Register of Causes of Death. Since 1994, causes of death have been recorded with ICD-10 codes [18]. Information on date of birth, sex, date of death, and migration status was obtained from the Danish Civil Registration System [19].

## Study population

The DNPR was used to identify all patients ≥18 years of age with a first-time primary or secondary hospital inpatient or outpatient clinic discharge diagnosis of atrial fibrillation or flutter, for which DOAC treatment was started between September 2011 and December 2015. Diagnoses of atrial fibrillation or flutter were obtained according to the ICD-10 coding system (S1 Table). The positive predictive value of atrial fibrillation or flutter in the DNPR is 93% [20].

Patients with atrial fibrillation or flutter were linked to the Danish National Prescription Registry to construct a cohort of patients who initiated DOAC treatment (S2 Table). September 2011 was chosen as the start of the study period, because rivaroxaban and dabigatran were then approved for stroke prevention in patients with atrial fibrillation by the European Medicines Agency [21–23]. Patients were included only if DOAC treatment was started within 3 months before or after the first atrial fibrillation diagnosis. This 3-month window is used since the atrial fibrillation diagnosis in outpatients is registered somewhere between the first and last visit date. Therefore, DOAC treatment could have been started before the diagnosis was registered.

The index date of the study and the start of follow-up was the date of DOAC treatment initiation. Outpatients with a secondary diagnosis of atrial fibrillation and more than 1 year between the admission and discharge dates were excluded, because the exact date of atrial fibrillation diagnosis within this period could not be assessed in the DNPR. Patients with a history of cancer other than nonmelanoma skin cancer before the index date were also excluded. Patients with prior short-term DOAC use for other indications, e.g., deep vein thrombosis, were not excluded. Two cohorts were compared: patients initiating factor Xa inhibitor (rivaroxaban, apixaban, or edoxaban) treatment and those initiating treatment with the direct thrombin inhibitor dabigatran.

## Study outcomes and follow-up

The primary outcome was a first diagnosis of any type of cancer other than nonmelanoma skin cancer. Secondary outcomes included individual cancer types and groups of cancer, including obesity-related cancers, smoking- and alcohol-related cancers, hematological cancers, immune-related cancers, neurological cancers, hormone-related cancers, and other cancers (S3 Table) [24–26]. Other secondary outcomes were metastatic disease at diagnosis, overall mortality, and cancer-specific mortality. Gastrointestinal bleeding was also included as a secondary outcome because bleeding from the gastrointestinal tract after DOAC initiation may prompt suspicion of gastrointestinal cancer and frequently leads to additional testing (e.g., endoscopy). Therefore, differences in cancer outcomes might result from diagnosis suspicion bias if a substantial difference in gastrointestinal bleeding risk exists between DOAC cohorts [27].

Cancer diagnoses were collected from the Danish Cancer Registry according to ICD-10 codes. All cancers were defined except for nonmelanoma skin cancer (ICD-10 code C44). Cancer mortality was collected from the Danish Register of Causes of Death according to ICD-10 codes. Gastrointestinal bleeding was assessed according to ICD-10 codes in the DNPR (S1 Table).

Patients were followed from the date of the DOAC initiation until the analyzed outcome event, death, emigration, loss to follow-up, end of study (December 31, 2020), or 5 years of follow-up, whichever came first. There was virtually complete follow-up in all patients; however, in very few cases (<10), we cannot follow the patients until death, emigration, or study end. An intention-to-treat approach was used in the main analyses, in which crossover or DOAC discontinuation was not taken into account. Additionally, in several sensitivity analyses, we assessed different outcomes in an on-treatment analysis and a time-varying analysis.

## Covariates

Data on the following comorbidities and comedications at baseline were collected: myocardial infarction, congestive heart failure, ischemic stroke, chronic obstructive pulmonary disease, liver disease, renal disease, inflammatory bowel disease, pancreatitis, gallstones, diabetes mellitus (including use of diabetes medication), hypertension (including use of antihypertensive agents), anemia, rheumatoid arthritis, alcohol dependency (including drugs for alcohol dependency), obesity and obesity-related disorders, platelet aggregation inhibitors, antihypertensive agents, lipid-lowering drugs, glucocorticoids, nonsteroidal anti-inflammatory drugs (NSAIDs), strong analgesics, and antidepressants (S1 and S2 Tables). Use of comedication was recorded if patients had a prescription in the Danish National Prescription Registry within 3 months before the index date.

## Statistical analysis

We characterized the study cohorts according to sex, age group at index date, calendar period of index date, comorbidities, and use of comedications. We performed propensity score weighting, using average treatment effect in the population weights, to compare the clinical outcomes of patients receiving factor Xa inhibitors with those receiving dabigatran. Propensity scores were computed with a multivariable logistic regression model with DOAC type as the dependent variable (factor Xa inhibitors versus dabigatran), including the following covariates: age; sex; myocardial infarction; congestive heart failure; ischemic stroke; chronic obstructive pulmonary disease; liver disease; renal disease; inflammatory bowel disease; pancreatitis; gallstones; diabetes mellitus (including use of diabetes medication); hypertension; anemia; rheumatoid arthritis; alcohol dependency (including drugs for alcohol dependency); obesity and obesity-related disorders; and use of platelet aggregation inhibitors, antihypertensive agents, lipid-lowering drugs, glucocorticoids, NSAIDs, strong analgesics, and antidepressants (S1 and S2 Tables). The positivity assumption (i.e., that any patient must have a nonzero probability of receiving either treatment) was supported by the observation that none of the individual weights were considered extreme (across all analyses: minimum weight 1.1; maximum weight 5.8; median weight 1.9, interquartile range [IQR] 1.7,2.2). Covariate balance after weighting was assessed according to standardized differences (S1 Fig).

Using the propensity score weighted cohorts, we constructed cumulative incidence curves for the outcomes of cancer, all-cause mortality, and gastrointestinal bleeding, by calculating the 5-year cumulative incidences, comparing the factor Xa inhibitor cohort with patients treated with dabigatran, using the Aalen–Johansen estimator, which accounts for the competing risk of death. For the primary and secondary outcomes, we used inverse probability of treatment weighted (IPTW) Fine and Gray competing risk regression models to calculate subdistribution hazard ratios (SHRs), with overall death as a competing event, by using a robust sandwich estimator to calculate the 95% confidence intervals (CIs). We assessed the proportional hazards assumption through visual inspection of log-minus-log plots in the populations,

weighted by their propensity scores, and found no major violations of the assumption. A visual distribution of propensity scores in both groups is presented in S2 Fig.

To evaluate the robustness of our estimates, we performed several sensitivity analyses. First, we assessed the association between DOAC treatments and different outcomes in the intention-to-treat analysis, adjusted for calendar year, because the proportion of patients with atrial fibrillation or flutter prescribed a factor Xa inhibitor increased each year during the study period. Next, we assessed outcomes in both cohorts in an on-treatment analysis and by using a time-varying exposure approach. In the on-treatment analysis, the patients were censored on the date at which the initial DOAC had not been prescribed for >200 days. In the time-varying analysis, DOAC use was assessed as a time-varying exposure. The exposure period was defined as the time from the index date until a switch from the factor Xa inhibitor cohort to the dabigatran cohort, or vice versa. In this analysis, exposure was not stopped after the DOAC had not been prescribed for >200 days. Additionally, we assessed the association between DOAC treatment initiation and outcomes in cause-specific Cox proportional hazards regression models.

After peer review of the manuscript, we added 2 sensitivity analysis. We performed an additional sensitivity analysis with follow-up time extended to 9 years to increase exposure time to the study drug. Of note, in this analysis, the mean follow-up time can differ between both cohorts since patients included in 2015 can only reach a maximum of 5 years of follow-up since we collected data up until 2020. We also performed a sensitivity analysis in which the inclusion period was stopped at the end of 2014 and all patients were followed for 5 years. In this analysis, the last included patient was followed until the end of 2019, excluding 2020 from the analysis, during which the Coronavirus Disease 2019 (COVID-19) pandemic started, resulting in less cancer diagnosis and higher mortality risks than previous years [28].

All analyses were performed in SAS version 9.4 (SAS Institute, Cary, NC, USA).

According to Danish legislation, registry-based research does not require ethical approval and informed consent but only permission from the Danish Data Protection Board. This study was approved by the Danish Data Protection Agency through registration at Aarhus University (Record Number 2016-051-000001/812). This study is reported as per the Strengthening the Reporting of Observational Studies in Epidemiology (STROBE) guideline (S1 STROBE Checklist).

## Results

A total of 23,712 patients who started DOAC treatment for newly diagnosed atrial fibrillation or flutter between 2011 and 2015 were included. Of these patients, 11,742 (49.5%) received a factor Xa inhibitor, and 11,970 received dabigatran (50.5%; Table 1). Before IPTW, patients in the factor Xa inhibitor cohort were older. The mean age was 75.2 years (standard deviation [SD] 11.2) in the factor Xa cohort and 71.7 years (SD 11.1) in the dabigatran cohort. Patients on factor Xa inhibitors were more often female (5,613/11,742 [47.8%] versus 4,953/11,970 [41.4%]), were less often included in the period between 2011 and 2013 (1,836/11,742, 15.6% versus 7,340/11,970; 61.3%), and had more comorbidities than those in the dabigatran cohort (Table 1).

The median duration of treatment was 2.95 (IQR 0.69,5.00) years for a factor Xa inhibitor, as compared with 2.13 (IQR 0.49,5.00) years for dabigatran. During the 5-year follow-up period, 1,058/11,742 (9.0%) patients in the factor Xa inhibitor cohort and 1,145/11,970 (9.6%) patients in the dabigatran cohort were diagnosed with cancer. After IPTW, the propensity score-weighted cumulative incidence of cancer was not substantially lower in the factor Xa inhibitor cohort (2,157/23,711; 9.11%, 95% CI [8.61%,9.63%]) than in the dabigatran cohort (2,294/23,715; 9.68%, 95% CI [9.14%,10.25%]; SHR 0.94, 95% CI 0.89 to 1.00; Fig 1). In

**Table 1. Baseline characteristics.**

| | Overall cohorts | | Propensity score-weighted cohorts | |
|---|---|---|---|---|
| | Factor Xa inhibitors N (%) | Thrombin inhibitors N (%) | Factor Xa inhibitors N (%) | Thrombin inhibitors N (%) |
| **Total** | **11,742 (100.0)** | **11,970 (100.0)** | **23,711 (100.0)** | **23,715 (100.0)** |
| Sex (female) | 5,613 (47.8) | 4,953 (41.4) | 10,576 (44.6) | 10,578 (44.6) |
| Age, years | | | | |
| <60 | 996 (8.5) | 1,560 (13.0) | 2,547 (10.7) | 2,552 (10.8) |
| 60–69 | 2,434 (20.7) | 3,365 (28.1) | 5,811 (24.5) | 5,805 (24.5) |
| 70–79 | 3,960 (33.7) | 3,987 (33.3) | 7,946 (33.5) | 7,952 (33.5) |
| 80+ | 4,352 (37.1) | 3,058 (25.5) | 7,406 (31.2) | 7,406 (31.2) |
| Years of index date | | | | |
| 2011–2013 | 1,836 (15.6) | 7,340 (61.3) | 3,634 (15.3) | 14,592 (61.5) |
| 2014–2015 | 9,906 (84.4) | 4,630 (38.7) | 20,077 (84.7) | 9,123 (38.5) |
| Myocardial infarction | 1,138 (9.7) | 1,025 (8.6) | 2,156 (9.1) | 2,164 (9.1) |
| Heart failure | 1,509 (12.9) | 1,267 (10.6) | 2,780 (11.7) | 2,783 (11.7) |
| Ischemic stroke | 1,626 (13.8) | 1,100 (9.2) | 2,733 (11.5) | 2,742 (11.6) |
| Hypertension | 9,846 (83.9) | 9,615 (80.3) | 19,473 (82.1) | 19,475 (82.1) |
| Anemia | 822 (7.0) | 522 (4.4) | 1,344 (5.7) | 1,344 (5.7) |
| COPD | 1,448 (12.3) | 1,104 (9.2) | 2,558 (10.8) | 2,571 (10.8) |
| Diabetes | 1,971 (16.8) | 1,754 (14.7) | 3,726 (15.7) | 3,723 (15.7) |
| Liver disease | 162 (1.4) | 147 (1.2) | 310 (1.3) | 311 (1.3) |
| Renal insufficiency | 427 (3.6) | 220 (1.8) | 647 (2.7) | 648 (2.7) |
| Inflammatory bowel disease | 403 (3.4) | 282 (2.4) | 684 (2.9) | 678 (2.9) |
| Pancreatitis | 158 (1.3) | 122 (1.0) | 282 (1.2) | 284 (1.2) |
| Gallstones | 853 (7.3) | 750 (6.3) | 1,615 (6.8) | 1,611 (6.8) |
| Rheumatoid arthritis | 291 (2.5) | 231 (1.9) | 523 (2.2) | 520 (2.2) |
| Alcohol dependency | 766 (6.5) | 739 (6.2) | 1,504 (6.3) | 1,502 (6.3) |
| Obesity | 887 (7.6) | 798 (6.7) | 1,687 (7.1) | 1,685 (7.1) |
| **Drugs used within 3 months before index date** | | | | |
| Antiplatelet therapy | 3,582 (30.5) | 3,298 (27.6) | 6,863 (28.9) | 6,858 (28.9) |
| Lipid-lowering therapy | 3,171 (27.0) | 3,208 (26.8) | 6,399 (27.0) | 6,391 (27.0) |
| NSAIDs | 1,217 (10.4) | 1,328 (11.1) | 2,560 (10.8) | 2,553 (10.8) |
| Corticosteroids | 756 (6.4) | 663 (5.5) | 1,429 (6.0) | 1,429 (6.0) |
| Strong analgesics | 1,629 (13.9) | 1,375 (11.5) | 2,998 (12.6) | 2,999 (12.7) |
| Antidepressants | 1,236 (10.5) | 1,059 (8.8) | 2,292 (9.7) | 2,294 (9.7) |

COPD, chronic obstructive pulmonary disease; NSAIDs, nonsteroidal anti-inflammatory drugs.

sensitivity analyses, we found no difference in cancer incidence in the intention-to-treat analysis adjusted for calendar year (SHR 0.94, 95% CI 0.88 to 1.00), the on-treatment analysis (SHR 0.99; 95% CI 0.92 to 1.06), or the time-varying analysis (HR 1.05; 95% CI 0.95 to 1.15) (S4 and S5 Tables).

Cancer stage was recorded in 1,302/2,203 (59.1%) of all patients with a cancer diagnosis during follow-up. In these patients, no difference in IPT-weighted cumulative incidence was observed in metastatic disease at diagnosis in the factor Xa inhibitor group (432/23,711; 1.86%, 95% CI [1.61%,2.16%]) and the dabigatran group (414/23,715; 1.79%, 95% CI [1.56%,2.05%]; SHR 1.04, 95% CI [0.91,1.19]). Moreover, no difference was observed in the incidence of grouped cancers, except for smoking- and alcohol-related cancers, whose incidence was lower

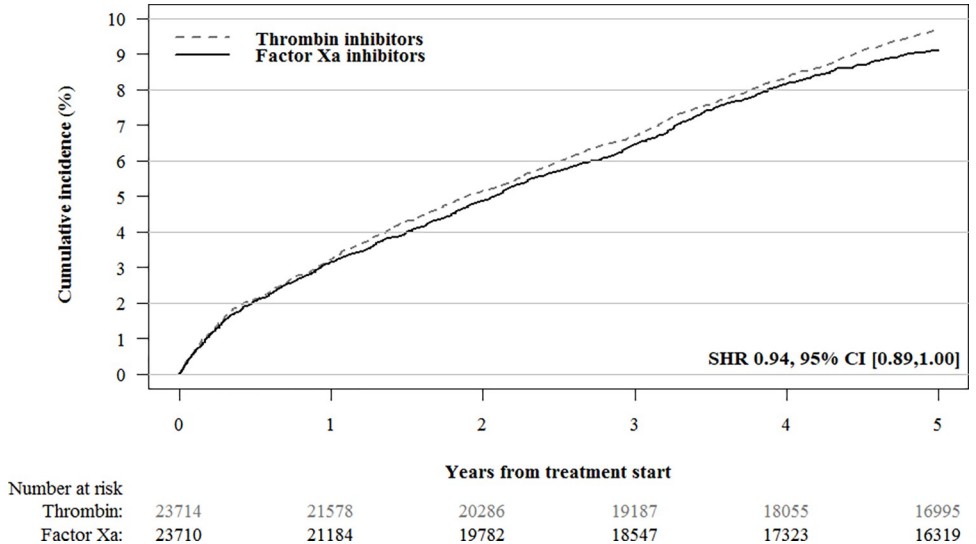

**Fig 1. IPT-weighted cumulative cancer incidence in patients treated with factor Xa inhibitors versus thrombin inhibitors.** The following variables were used to compute propensity scores and construct propensity score weighted cohorts: age; sex; myocardial infarction; congestive heart failure; ischemic stroke; chronic obstructive pulmonary disease; liver disease; renal disease; inflammatory bowel disease; pancreatitis; gallstones; diabetes mellitus (including use of diabetes medication); hypertension; anemia; rheumatoid arthritis; alcohol dependency (including drugs for alcohol dependency); obesity and obesity-related disorders; and use of platelet aggregation inhibitors, antihypertensive agents, lipid-lowering drugs, glucocorticoids, NSAIDs, strong analgesics, and antidepressants. IPT, inverse probability of treatment; NSAID, nonsteroidal anti-inflammatory drug.

in the factor Xa inhibitor cohort (519/23,711; 2.24%, 95% CI [1.97%,2.53%]) than the dabigatran cohort (624/23,715; 2.69%, 95% CI [2.41%,3.00%]; SHR 0.83, 95% CI [0.74,0.93]; Table 2) For individual tumor types, the cumulative incidence was lower in the factor Xa inhibitor cohort than the dabigatran cohort for lung cancer (307/23,711; 1.32%, 95% CI [1.12,1.56] versus 364/23,715; 1.57%, 95% CI [1.36,1.82]; SHR 0.84, 95% CI [0.72,0.98]), hematological cancer (181/23,711; 0.78%, 95% CI [0.64,0.97] versus 276/23,715; 1.19%, 95% CI [1.01,1.41]; SHR 0.66, 95% CI [0.55,0.79]), and gastroesophageal cancer (64/23,711; 0.28%, 95% CI [0.19,0.40] versus 103/23,715; 0.45%, 95% CI [0.33,0.61]; SHR 0.63, 95% CI [0.46,0.85]), but not the other cancer types (Table 2). These findings were also observed in the sensitivity analysis of IPT-weighted SHR adjusted for calendar year, but in the on-treatment analysis, factor Xa inhibitors were only associated with lower incidence of smoking- and alcohol-related cancers and hematological cancers (S4 and S5 Tables). In the time-varying analysis, factor Xa inhibitor initiation was not associated with a lower risk of these outcomes.

During the 5-year follow-up period, 4,133/11,742 (35.2%) patients in the factor Xa inhibitor cohort and 2,881/11,970 (24.1%) patients in the dabigatran cohort died. After IPTW, the all-cause mortality was higher in the factor Xa inhibitor cohort than the dabigatran cohort (SHR 1.17, 95% CI [1.13,1.21]; Table 2 and Fig 2). No difference in cancer-specific mortality was observed between factor Xa inhibitors (1,028/23,711; 4.34%, 95% CI [4.02%,4.68%]) and dabigatran (1,001/23,715; 4.23%, 95% CI [3.83%,4.66%]; SHR 1.03, 95% CI [0.94,1.12]) treatments. Gastrointestinal bleeding occurred less frequently in the factor Xa inhibitor cohort (1,245/23,711; 5.26%, 95% CI [4.90%,5.64%]) than the dabigatran cohort (1,540/23,715; 6.50%, 95% CI [6.05%,6.98%]; SHR 0.80; 95% CI, 0.75,0.87; Table 2 and Fig 3).

In the sensitivity analysis with 9 years of follow-up, cancer incidence was significantly lower in the factor Xa inhibitor group than in the dabigatran group (2,277/23,711 (9.60%) versus 2,706/23,715 (11.41%); SHR 0.88, 95% CI [0.83,0.93]) (S6 Table). Importantly, in contrast to

**Table 2. IPT-weighted cumulative incidence and SHRs in the intention-to-treat analysis for factor Xa inhibitor cohort versus the dabigatran cohort during 5 years of follow-up.**

| Outcome | Factor Xa inhibitor cohort | | Dabigatran cohort | | IPT-weighted SHR (95% CI) | P value* |
|---|---|---|---|---|---|---|
| | N = 23,711 | | N = 23,715 | | | |
| | Total | % (95% CI) | Total | % (95% CI) | | |
| Cancer total | 2,157 | 9.11 (8.61,9.63) | 2,294 | 9.68 (9.14,10.25) | 0.94 (0.89,1.00) | 0.0357 |
| Metastatic disease at diagnosis† | 432 | 1.86 (1.61,2.16) | 414 | 1.79 (1.56,2.05) | 1.04 (0.91,1.19) | 0.5538 |
| Cancer-specific mortality | 1,028 | 4.34 (4.02,4.68) | 1,001 | 4.23 (3.83,4.66) | 1.03 (0.94,1.12) | 0.5389 |
| All-cause mortality | 7,416 | 31.31 (30.37,32.29) | 6,531 | 27.56 (26.69,28.45) | 1.17 (1.13,1.21) | < .0001 |
| Gastrointestinal bleeding | 1,245 | 5.26 (4.90,5.64) | 1,540 | 6.50 (6.05,6.98) | 0.80 (0.75,0.87) | < .0001 |
| **Cancer groups** | | | | | | |
| Obesity-related cancer | 642 | 2.75 (2.45,3.09) | 616 | 2.64 (2.38,2.93) | 1.04 (0.93,1.17) | 0.4537 |
| Hormone-related cancer | 528 | 2.27 (2.02,2.54) | 516 | 2.22 (1.96,2.51) | 1.02 (0.91,1.15) | 0.7235 |
| Smoking- and alcohol-related cancers | 519 | 2.24 (1.97,2.53) | 624 | 2.69 (2.41,3.00) | 0.83 (0.74,0.93) | 0.0019 |
| Immune-related cancer | 140 | 0.61 (0.49,0.74) | 119 | 0.52 (0.40,0.67) | 1.18 (0.92,1.50) | 0.1932 |
| Neurological cancer | 87 | 0.38 (0.28,0.50) | 85 | 0.37 (0.28,0.48) | 1.03 (0.76,1.38) | 0.8628 |
| Other cancers | 60 | 0.26 (0.17,0.38) | 58 | 0.25 (0.17,0.37) | 1.01 (0.70,1.45) | 0.9583 |
| **Cancer types** | | | | | | |
| Colorectal | 362 | 1.56 (1.37,1.77) | 376 | 1.61 (1.40,1.85) | 0.96 (0.83,1.11) | 0.5854 |
| Lung | 307 | 1.32 (1.12,1.56) | 364 | 1.57 (1.36,1.82) | 0.84 (0.72,0.98) | 0.0276 |
| Prostate | 296 | 1.28 (1.06,1.53) | 293 | 1.26 (1.06,1.50) | 1.01 (0.86,1.19) | 0.8963 |
| Breast | 202 | 0.87 (0.73,1.05) | 200 | 0.86 (0.72,1.04) | 1.01 (0.83,1.23) | 0.9171 |
| Hematological | 181 | 0.78 (0.64,0.97) | 276 | 1.19 (1.01,1.41) | 0.66 (0.55,0.79) | < .0001 |
| Urogenital | 145 | 0.63 (0.51,0.77) | 134 | 0.58 (0.45,0.75) | 1.09 (0.86,1.38) | 0.4827 |
| Gynecological | 89 | 0.38 (0.29,0.50) | 70 | 0.30 (0.21,0.43) | 1.26 (0.92,1.73) | 0.1463 |
| Gastroesophageal | 64 | 0.28 (0.19,0.40) | 103 | 0.45 (0.33,0.61) | 0.63 (0.46,0.85) | 0.0031 |
| Hepatobiliary | 29 | 0.13 (0.08,0.21) | 38 | 0.17 (0.11,0.25) | 0.77 (0.47,1.24) | 0.2771 |
| Brain | 17 | 0.07 (0.04,0.14) | 26 | 0.11 (0.06,0.19) | 0.66 (0.36,1.22) | 0.1869 |

CI, confidence interval; HR, hazard ratio; IPT, inverse probability of treatment; SHR, subdistribution hazard ratio.

*Chi-squared tests have been used to determine P values.

†Disease stage was recorded in 59.1% of patients with cancer during follow-up.

the other analyses, the mean follow-up time was higher in the dabigatran group (5.98 years (SD 2.40)) than in the factor Xa inhibitor group (4.60 years (SD 2.14)). In the sensitivity analysis excluding the follow-up during the COVID-19 pandemic, there was no difference in cancer between the 2 cohorts (1,547/15,772 (9.81%) versus 1,516/15,801 (9.59%); SHR 1.02, 95% CI [0.95,1.10]) (S7 Table).

Results for all sensitivity analyses are presented in S6 and S7 Tables.

## Discussion

In this cohort study of patients with atrial fibrillation or flutter initiating DOAC treatment, we sought to investigate whether the use of factor Xa inhibitor had an effect on cancer incidence and cancer-related mortality. After propensity score weighting, we found no substantial difference in incidence of cancer or cancer-related mortality between patients taking factor Xa inhibitors and those taking dabigatran. Although the upper bound of the 95% CI of the primary analysis was 1.00, suggesting a possible effect, no difference was found in any of the sensitivity analysis, including an analysis in which we excluded the first year of COVID-19 during

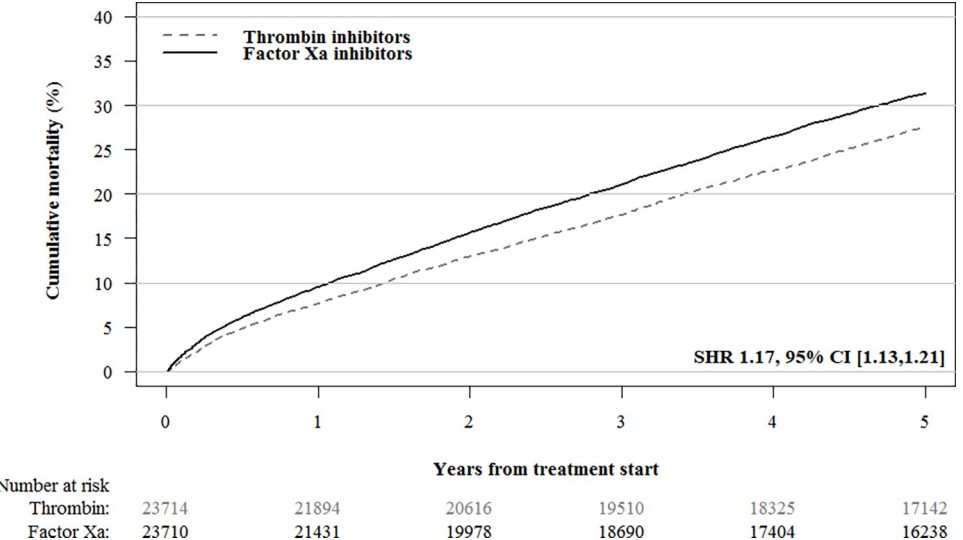

**Fig 2. IPT-weighted cumulative incidence curves for mortality in patients treated with factor Xa inhibitors versus thrombin inhibitors.** The following variables were used to compute propensity scores and construct propensity score weighted cohorts: age; sex; myocardial infarction; congestive heart failure; ischemic stroke; chronic obstructive pulmonary disease; liver disease; renal disease; inflammatory bowel disease; pancreatitis; gallstones; diabetes mellitus (including use of diabetes medication); hypertension; anemia; rheumatoid arthritis; alcohol dependency (including drugs for alcohol dependency); obesity and obesity-related disorders; and use of platelet aggregation inhibitors, antihypertensive agents, lipid-lowering drugs, glucocorticoids, NSAIDs, strong analgesics, and antidepressants. IPT, inverse probability of treatment; NSAID, nonsteroidal anti-inflammatory drug.

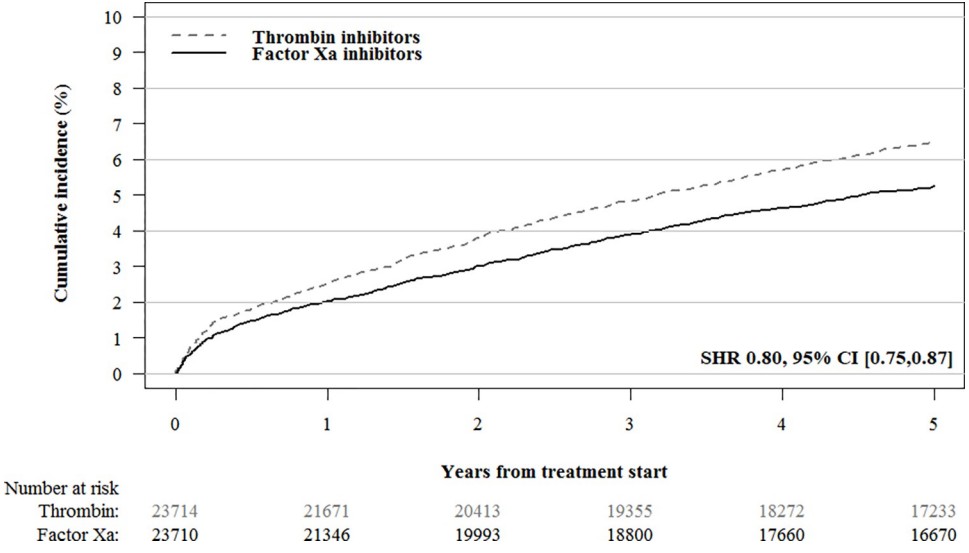

**Fig 3. IPT-weighted cumulative gastrointestinal bleeding incidence in patients treated with factor Xa inhibitors versus thrombin inhibitors.** The following variables were used to compute propensity scores and construct propensity score weighted cohorts: age; sex; myocardial infarction; congestive heart failure; ischemic stroke; chronic obstructive pulmonary disease; liver disease; renal disease; inflammatory bowel disease; pancreatitis; gallstones; diabetes mellitus (including use of diabetes medication); hypertension; anemia; rheumatoid arthritis; alcohol dependency (including drugs for alcohol dependency); obesity and obesity-related disorders; and use of platelet aggregation inhibitors, antihypertensive agents, lipid-lowering drugs, glucocorticoids, NSAIDs, strong analgesics, and antidepressants. IPT, inverse probability of treatment; NSAID, nonsteroidal anti-inflammatory drug.

which fewer cancer diagnosis were made, possibly resulting in fewer cancer incidences in the factor Xa inhibitor group.

A preclinical study in mice by Graf and colleagues has shown that factor X is a crucial driver of important innate immune signaling pathways [13]. The authors assessed the effects of factor X and rivaroxaban on tumor burden and metastasis in breast cancer and fibrosarcoma. Whether inhibition of factor X also works in humans or in other cancer types is unclear. Mechanisms of innate immune evasion are different across tumor types and tumors respond differently to immunotherapy [29]. Therefore, based on this hypothesis, differences in response to factor Xa inhibition across tumor types are expected. In the present cohort, we found no evidence for an effect of factor Xa inhibitors on overall cancer occurrence. We did find an association with factor Xa inhibitor use and lower incidence of alcohol- and smoking-related cancers, lung cancer, hematological cancer, and gastroesophageal cancer, suggesting that factor Xa may play a role in cancer progression in specific cancer types. However, these findings on secondary outcomes should be interpreted with caution. Precision in certain cancer-specific estimates was low due to low event numbers. As for the main analysis, potential residual confounding precludes strong conclusions about causality. Yet, as patients with factor Xa inhibitors more often had comorbidities, the incidence of alcohol- and smoking-related cancers and lung cancer would likely have been higher in this group instead of lower. Finally, differences in subgroup outcomes might have been mediated by differences in bleeding risk between the cohorts. For example, gastrointestinal bleeding occurs more frequently in patients using dabigatran, thus potentially leading to an earlier diagnosis of gastrointestinal cancer [30]. Not all subgroup analysis showed the same results in sensitivity analysis. The time-varying results might differ from the intention-to-treat and on-treatment analyses, given that in time-varying analyses, we have not used the IPTW approach as they were directly adjusted for confounding variables, resulting in a decrease in the precision of the estimates. In the on-treatment analysis, patients were followed for 200 days after last subscription, resulting in a shorter overall follow-up time. Although the intention-to-treat and on-treatment analyses show largely the same results, shorter follow-up time could have an effect on outcomes.

Several limitations of the present study merit consideration. First, we chose dabigatran as the comparator, assuming no preference for a particular type of DOAC, because all DOACs are usually considered equally efficacious and safe in patients with atrial fibrillation [21,31]. However, considerable differences in baseline characteristics and all-cause mortality risk were observed between the cohorts. Whereas dabigatran use was more prevalent at the beginning of the study period, because this drug was licensed first, factor Xa inhibitors were increasingly prescribed later, after clinicians had been assured of their efficacy and safety profiles. Factor Xa inhibitors also seemed to be prescribed preferentially in older and frail patients [32]. Although we used IPTW to decrease confounding, the all-cause mortality risk was still higher in the factor Xa cohort than in the dabigatran cohort, which may be the result of residual confounding. Guidelines suggest to be cautious when prescribing dabigatran to elderly patients because of concerns of bleeding and myocardial infarction [33–35]. Factor Xa inhibitors are also preferred over dabigatran in patients with chronic kidney disease [34]. Nonetheless, previous observational studies suggest that factor Xa inhibitors do not increase mortality compared to dabigatran [36]. The sensitivity analysis excluding the time period of COVID-19, during which factor Xa use was more prevalent, suggests that this pandemic may also have led to a higher mortality risk in the factor Xa group [28]. Therefore, we performed a sensitivity analysis with 5 years follow-up and the inclusion period ending at the end of 2019, excluding overlap with the first year of the COVID-19 pandemic. In this analysis, the point estimate of SHR for cancer was slightly higher compared to the primary analysis, suggesting a small potential effect of COVID-19, although estimates were largely comparable between both analyses. Second, a

recent study has suggested that dabigatran prevents cancer spread in mice with colon cancer by obliterating transforming growth factor-β1 (TGF-β1), an immunosuppressive cytokine with important roles in oncogenesis; therefore, we could not rule out competing effects on cancer progression [37]. Third, the timeframe in which a positive effect of factor Xa inhibitors on cancer occurrence or progression could be expected is unknown. Because most cancers develop over many years, our relatively short follow-up period might not have been sufficiently long to detect differences in cancer incidence and mortality [38]. Nonetheless, the preclinical study in mice by Graf and colleagues on which the hypothesis is based [13] suggests that factor Xa inhibitors should prevent progression of cancer and cancer growth with a mechanism similar to PD-L1 inhibition, which suggests it could also have an effect when a tumor has not yet been diagnosed but already has had several years since inception, possibly resulting in postponement of a tumor becoming clinically apparent. We therefore hypothesized that 5 years would be sufficient to observe a potential effect. To increase exposure time to the study drug, we performed a sensitivity analysis with follow-up time increased to 9 years. In this analysis, we did find a lower risk of cancer in the factor Xa inhibitor group, but since the majority of patients on dabigatran were included in the beginning of the study period, important differences in follow-up time were observed. Therefore, no clear conclusions can be drawn based on this analysis. Together, these potential biases might have led to an underestimation of the relationship between factor Xa inhibitors and cancer, and we therefore cannot rule out a potential causal link.

Transferability from animal models to humans could also be a problem when testing this hypothesis. While the importance of factor X in cancer progression was proved in preclinical mice studies, it is unknown whether factor X and factor Xa inhibition plays a similar role in humans and different cancer types. Previous studies evaluating the association between DOACs and cancer either have combined thrombin inhibitors with factor Xa inhibitors [39] or have assessed only cancer mortality in patients who were already diagnosed with cancer or had a relatively short follow-up [40]. Therefore, large studies are necessary to corroborate these findings in various cancer types. Because RCTs to address this question are not feasible, given the large number of patients and long-term follow-up required, large cohort studies can serve as an alternative, because they provide detailed healthcare data on medication prescription, medical history, comedication, and causes of death. In this study, we assessed the incidence of onset of many cancer types in patients with atrial fibrillation treated with factor Xa inhibitors or thrombin inhibitors, with 5 years' follow-up. Alternatively, long-term follow-up data from Phase III trials comparing DOACs to vitamin K antagonists for atrial fibrillation could be used to assess cancer incidence in different treatment arms [22,23,41].

In conclusion, we did not observe lower cancer incidence in patients with atrial fibrillation taking a factor Xa inhibitor rather than dabigatran during a 5-year follow-up. Our data do not support the hypothesis that factor Xa inhibition strongly limits cancer growth overall or reduces cancer-related mortality.

## Supporting information

**S1 STROBE Checklist. Statement Checklist of items that should be included in reports of cohort studies.** *Give information separately for exposed and unexposed groups. **Note:** An Explanation and Elaboration article discusses each checklist item and gives methodological background and published examples of transparent reporting. The STROBE checklist is best used in conjunction with this article (freely available on the websites of PLoS Medicine at http://www.plosmedicine.org/, Annals of Internal Medicine at http://www.annals.org/, and Epidemiology at http://www.epidem.com/). Information on the STROBE Initiative is available

at http://www.strobe-statement.org.
(DOCX)

**S1 Table. Index variable, covariates, and noncancer-related outcomes from the DNPR.** ICD-8, 8th revision of the International Statistical Classification of Diseases and Related Health Problems; ICD-10, 10th revision of the International Statistical Classification of Diseases and Related Health Problems.
(DOCX)

**S2 Table. Drugs and ATC codes.** *Used to define diagnosis. ATC, Anatomical Therapeutic Chemical; NSAIDs, nonsteroidal anti-inflammatory drugs.
(DOCX)

**S3 Table. Table of cancer diagnosis and cancer groups.** ICD-10, 10th revision of the International Statistical Classification of Diseases and Related Health Problems.
(DOCX)

**S4 Table. SHRs adjusted for calendar year (intention-to-treat), cause-specific HR (intention-to-treat analysis), and adjusted HRs (time-varying analysis) for different outcomes in the factor Xa inhibitor versus dabigatran cohorts.** CI, confidence interval; HR, hazard ratio; IPT, inverse probability of treatment; SHR, subdistribution hazard ratio. *Disease stage was recorded in 59.1% of patients with cancer during follow-up.
(DOCX)

**S5 Table. SHRs in the on-treatment analysis for different outcomes in the factor Xa inhibitor versus dabigatran cohorts.** CI, confidence interval; HR, hazard ratio; IPT, inverse probability of treatment; SHR, subdistribution hazard ratio. *Disease stage was recorded in 59.1% of patients with cancer during follow-up.
(DOCX)

**S6 Table. Sensitivity analysis with IPT-weighted cumulative incidence and SHRs for different outcomes in the factor Xa inhibitor cohort versus the dabigatran cohort during 9 years of follow-up.** CI, confidence interval; HR, hazard ratio; IPT, inverse probability of treatment; SHR, subdistribution hazard ratio.
(DOCX)

**S7 Table. Sensitivity analysis with IPT-weighted cumulative incidence and SHRs for different outcomes in the factor Xa inhibitor cohort versus the dabigatran cohort during 5 years of follow-up with inclusion period between 2011 and 2014.** CI, confidence interval; HR, hazard ratio; IPT, inverse probability of treatment; SHR, subdistribution hazard ratio.
(DOCX)

**S1 Fig. Standardized differences for covariates included in the propensity score model.** CHF, congestive heart failure; COPD, chronic obstructive pulmonary disease; IBD, inflammatory bowel disease; Istroke, ischemic stroke; MI, myocardial infarction; NSAIDs, nonsteroidal anti-inflammatory drugs; prop score, propensity score; obs, observations; w3m, within 3 months.
(DOCX)

**S2 Fig. Propensity score distribution in the factor Xa inhibitor and dabigatran cohorts.** Bars represent histogram of propensity scores in the 2 cohorts (expressed as percentages per 100) and lines represent kernel density plots of propensity scores prior to inverse probability of treatment weighting.
(DOCX)

**S1 Appendix. Supporting information.**
(DOCX)

## Author Contributions

**Conceptualization:** Floris Bosch, Suzanne C. Cannegieter, Nick van Es.

**Data curation:** Erzsébet Horváth-Puhó.

**Formal analysis:** Erzsébet Horváth-Puhó.

**Methodology:** Floris Bosch, Erzsébet Horváth-Puhó, Suzanne C. Cannegieter, Nick van Es.

**Resources:** Henrik T. Sørensen.

**Supervision:** Henrik T. Sørensen.

**Writing – original draft:** Floris Bosch, Erzsébet Horváth-Puhó.

**Writing – review & editing:** Suzanne C. Cannegieter, Nick van Es, Henrik T. Sørensen.

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
