## [Editor Report · Decision Letter 0]

22 Nov 2023

Dear Dr Bosch, 

Thank you for submitting your manuscript entitled "Direct factor Xa inhibitors and the risk of cancer and cancer mortality: a population-based cohort study" for consideration by PLOS Medicine.

Your manuscript has now been evaluated by the PLOS Medicine editorial staff and I am writing to let you know that we would like to send your submission out for external peer review.

Please re-submit your manuscript within two working days, i.e. by Nov 24 2023 11:59PM.

Feel free to email me at aschaefer@plos.org if you have any queries relating to your submission.

Kind regards,

Alexandra Schaefer, PhD

Associate Editor

PLOS Medicine

---

## [Decision Letter · Decision Letter 1]

5 Jan 2024

Dear Dr. Bosch,

Thank you very much for submitting your manuscript "Direct factor Xa inhibitors and the risk of cancer and cancer mortality: a population-based cohort study" (PMEDICINE-D-23-03433R1) for consideration at PLOS Medicine. 

Your paper was evaluated by an associate editor and discussed among all the editors here. It was also discussed with an academic editor with relevant expertise, and sent to independent reviewers, including a statistical reviewer. The reviews are appended at the bottom of this email and any accompanying reviewer attachments can be seen via the link below:

[LINK]

In light of these reviews, I am afraid that we will not be able to accept the manuscript for publication in the journal in its current form, but we would like to consider a revised version that addresses the reviewers' and editors' comments. Obviously we cannot make any decision about publication until we have seen the revised manuscript and your response, and we plan to seek re-review by one or more of the reviewers. 

We expect to receive your revised manuscript by Jan 26 2024 11:59PM. Please email me (aschaefer@plos.org) if you have any questions or concerns.

We look forward to receiving your revised manuscript. 

Sincerely,

Alexandra Schaefer, PhD

PLOS Medicine

plosmedicine.org

ACADEMIC EDITOR COMMENTS

My review of this paper aligns most closely with Reviewer 1. This is a well-designed, well-executed negative study. The sentence in the abstract, discussion, and cover letter "subgroup analyses suggested that it might decrease the risk of several cancer types" is somewhat exaggerated -- there were three cancer types (out of 10) where cancer incidence was lower in the Factor Xa inhibitor group (Table 2), but these data did not consistently hold up in the Table S4 analysis.

EDITORIAL COMMENTS

The editors concur with the reviewers that the follow-up is rather short for the outcome of interest. Therefore, we suggest extending the follow-up period to provide a more robust assessment and agree with the statistical reviewer that a sensitivity analysis adjusting for age is merited. In addition, we ask that you report absolute numbers throughout the entire manuscript and revise your discussion to include a more formal discussion of limitations.

GENERAL COMMENTS

1) Please include page numbers and line numbers in the manuscript file. Use continuous line numbers (do not restart the numbering on each page). For review purposes, we started counting the Abstract as page 1.

2) Please cite the reference numbers in square brackets (e.g., “We used the techniques developed by our colleagues [19] to analyze the data”). Citations should be preceding punctuation.

3) Please cite your Supporting Information as outlined here: https://journals.plos.org/plosmedicine/s/supporting-information

FINANCIAL DISCLOSURE

Please enter a financial disclosure statement that describes the sources of funding for the work included in this submission. If the study received no specific funding, please enter “The author(s) received no specific funding for this work.”

COMPETING INTEREST

All authors must declare their relevant competing interests per the PLOS policy, which can be seen here: https://journals.plos.org/plosmedicine/s/competing-interests

Please add this statement to the manuscript's Competing Interests: "SC is an Academic Editor on PLOS Medicine's editorial board."

TITLE

Please include the study setting, i.e. Denmark, in the title.

ABSTRACT

1) PLOS Medicine requests that main results are quantified with 95% CIs as well as p values. When reporting p values please report as p<0.001 and where higher as the exact p value p=0.002, for example. For the purposes of transparent data reporting, if not including the aforementioned please clearly state the reasons why not.

2) Throughout, suggest reporting statistical information as follows to improve clarity for the reader “22% (95% CI [13%,28%]; p</=)”. Please amend throughout the abstract and main manuscript. Please note the use of commas to separate upper and lower bounds, as opposed to hyphens as these can be confused with reporting of negative values.

3) When a p value is given, please specify the statistical test used to determine it. 

4) Please structure your abstract using the PLOS Medicine headings (Background, Methods and Findings, Conclusions). Please combine the Methods and Findings sections into one section, “Methods and findings”.

5) Please explain what rivaroxaban is, e.g. “…rivaroxaban, a direct anticoagulant and factor Xa inhibitor,..” or similar.

6) Please ensure that all numbers presented in the abstract are present and identical to numbers presented in the main manuscript text.

7) Please include the age of the study population and the main outcome measures.

8) Please include the important dependent variables that are adjusted for in the analyses.

9) In the last sentence of the Abstract Methods and Findings section, please describe the main limitation(s) of the study's methodology.

AUTHOR SUMMARY

At this stage, we ask that you include a short, non-technical Author Summary of your research to make findings accessible to a wide audience that includes both scientists and non-scientists. The Author Summary should immediately follow the Abstract in your revised manuscript. This text is subject to editorial change and should be distinct from the scientific abstract. Please see our author guidelines for more information: https://journals.plos.org/plosmedicine/s/revising-your-manuscript#loc-author-summary.

The summary should include 2-3 single sentence, individual bullet points under each of the questions. The last bullet under ‘What Do These Findings Mean?’ point should describe the main limitation of the study's methodology.

It may be helpful to review currently published articles for examples which can be found on our website here https://journals.plos.org/plosmedicine/

INTRODUCTION

1) p.2: Please introduce the abbreviation “RCT” before its first use.

2) “Pharmacological blockade…immune checkpoint inhibition.” – please provide reference.

METHODS AND RESULTS

General points:

1) For all observational studies, in the manuscript text, please indicate: (1) the specific hypotheses you intended to test, (2) the analytical methods by which you planned to test them, (3) the analyses you actually performed, and (4) when reported analyses differ from those that were planned, transparent explanations for differences that affect the reliability of the study's results. If a reported analysis was performed based on an interesting but unanticipated pattern in the data, please be clear that the analysis was data-driven.

2) Did your study have a prospective protocol or analysis plan? Please state this (either way) early in the Methods section.

3) Please ensure that the study is reported according to the STROBE guideline and include the completed STROBE checklist as Supporting Information. When completing the checklist, please use section and paragraph numbers, rather than page numbers. Please add the following statement, or similar, to the Methods: ""This study is reported as per the Strengthening the Reporting of Observational Studies in Epidemiology (STROBE) guideline (S1 Checklist).""

4) PLOS Medicine requests that main results are quantified with 95% CIs as well as p values. When reporting p values please report as p<0.001 and where higher as the exact p value p=0.002, for example. For the purposes of transparent data reporting, if not including the aforementioned please clearly state the reasons why not. Please include any important dependent variables that are adjusted for in the analyses. We suggest reporting statistical information as detailed above – see under ABSTRACT.

5) Please present numerators and denominators for percentages, at least in the Tables [not necessarily each time they're mentioned].

6) Please define "lost to follow-up" as used in this study. Other reasons for exclusion should be defined.

7) p.6: Please define ‘NSAID’.

8) p.7: “The study was reported to the Danish Data Protection Board by Aarhus University” - Please confirm that this specific study was reviewed and approved by an institutional review board (ethics committee) before the study began and provide additional details regarding consent. Please provide the specific name of the ethics committee/IRB that approved your study. Once you have amended this/these statement(s) in the Methods section of the manuscript, please add the same text to the “Ethics Statement” field of the submission form.

9) p.7: “Of these patients, 11,742 (50%) received a factor Xa inhibitor, and 11,970 received dabigatran (50%; Table 1).” – To make the values closer to the actual distribution (49.5% and 50.5%), we suggest adding a decimal point to the percentages.

10) p.7: Please define ‘SD’ at first use.

11) p.7: “were more often female (47.8% vs 41.4%), were less often included in the period between 2011 and 2013 (15.6% vs 61.3%), and had more comorbidities than those in the dabigatran cohort (Table 1).” – please revise.

12) p.8: “After IPTW, the all-cause mortality was higher in the factor Xa inhibitor cohort than the dabigatran cohort (HR 1.17, 95% CI 1.13–1.21; Table 2 and Figure 2).” – Should HR be SHR here?

DISCUSSION

1) p.9 “A pre-clinical study in mice by Graf et al. has shown that factor Xa is a crucial driver of the innate immune signaling pathway.” – please provide the reference number.

2) p.10: Please define ‘FX’ or write in full.

3) p.10: Please define ‘VKA’ or write in full.

TABLES

1) Please define abbreviations used in each table (including those in Supporting Information files).

2) Please note the use of commas to separate upper and lower bounds, as opposed to hyphens as these can be confused with reporting of negative values.

3) Table 2: Please define ‘IPTW’, ‘CI’, ‘HR’.

FIGURES

1) For all Figures, please ensure that you have complied with our figures requirements http://journals.plos.org/plosmedicine/s/figures.

2) Please consider avoiding the use of red and green in order to make your figure more accessible to those with colour blindness. 

3) Please in the figure legend/description, define abbreviations used in each figure (e.g. ‘IPTW’; including those in Supporting Information files).

4) Please provide titles, legends and descriptions for all figures (including those in Supporting Information files).

SUPPLEMENTARY MATERIAL

1) For supplementary figures and tables, please see the general comments under TABLES and FIGURES (color, abbreviations, titles, descriptions, etc.) and amend accordingly.

2) We suggest reporting statistical information as detailed above – see under ABSTRACT. Please define all numerical values (e.g. Supplemental Table 4 does not specify the numerical value of the numbers in brackets).

3) Supplemental Figure 1: Please ensure that all covariates and their abbreviations are properly defined as well as the abbreviations used in the figure legend.

4) Supplemental Figure 2: Please ensure to reference Supplemental Figure 2 in the main manuscript. Please describe the meaning of the lines and bars and add a unit for ‘Percent’.

REFERENCES

1) PLOS uses the numbered citation (citation-sequence) method and first six authors, et al.

2) Please ensure that journal name abbreviations match those found in the National Center for Biotechnology Infor

---

## [Decision Letter · Decision Letter 2]

18 Mar 2024

Dear Dr. Bosch,

Thank you very much for re-submitting your manuscript "Direct factor Xa inhibitors and the risk of cancer and cancer mortality: a population-based cohort study" (PMEDICINE-D-23-03433R2) for review by PLOS Medicine.

Thank you for your detailed response to the editors' and reviewers' comments. I have discussed the paper with my colleagues and the academic editor, and it has also been seen again by two of the original reviewers. The changes made to the paper were satisfactory to the reviewers. As such, we intend to accept the paper for publication, pending your attention to the editorial comments below in a further revision. When submitting your revised paper, please once again include a detailed point-by-point response to the editorial comments.

[LINK]

In revising the manuscript for further consideration here, please ensure you address the specific points made by each reviewer and the editors. In your rebuttal letter you should indicate your response to the reviewers' and editors' comments and the changes you have made in the manuscript. Please submit a clean version of the paper as the main article file. A version with changes marked must also be uploaded as a marked up manuscript file. Please also check the guidelines for revised papers at http://journals.plos.org/plosmedicine/s/revising-your-manuscript for any that apply to your paper.

We ask that you submit your revision within 1 week (Mar 25 2024). However, if this deadline is not feasible, please contact me by email, and we can discuss a suitable alternative.

Please do not hesitate to contact me directly with any questions (aschaefer@plos.org). If you reply directly to this message, please be sure to 'Reply All' so your message comes directly to my inbox.

We look forward to receiving the revised manuscript.

Sincerely,

Alexandra Schaefer, PhD

Associate Editor

PLOS Medicine

plosmedicine.org

Requests from Editors:

ACADEMIC EDITOR COMMENTS

I think the authors have done a reasonable job of responding to the reviewers' comments.

I would like to note that this is a completely negative study, and furthermore, the Factor Xa inhibitors increased all-cause mortality compared to standard thrombin inhibitors.

I think the authors need to highlight the increase in all-cause mortality with factor Xa inhibitors in the conclusion of their abstract and in the "What did the researchers do and find?" and "What do these findings mean?" sections of the Author Summary. It is important that this important (and adverse) finding be clearly stated and obvious even to the quick reader.

EDITORIAL COMMENTS

Please be sure to include absolute numbers of events when reporting results, including in the sensitivity analyses, (e.g., #s of all-cause deaths, cancer-specific deaths, cancer subtypes) for the matched group analysis. In the text, it appears that you are reporting the total N for incident cancers (and deaths), but only for the overall group, not after matching, so the absolute numbers of events used for IPT-weighted cumulative incidence/SHR are not available in the text. Please also include the absolute numbers of events in the relevant tables, such as Table 2 (i.e., add two columns in table 2 to report numbers of events in each treatment group for each outcome listed) as well as Tables S4 and S5.

ABSTRACT

1) l.41: Please define ‘SD’ at first use. 

2) ll.46-47: Please include absolute numbers of events and denominators before percentages: e.g., (n/N; 9.11%, 95%CI [8.61%,9.63%]). Please do the same for lines 48-50 and when you report these data in the Results section (e.g., lines 252-253; 259-260, etc.).

3) Please state the main limitation at the end of the “Methods and Findings” section. Please also consider modifying this sentence as follows, “The main limitations of the study were the possibility of residual confounding and the short follow up period.”

AUTHOR SUMMARY

Thank for providing the Author Summary. We feel that the Author Summary in its current form does not provide sufficient detail (please also see the comments provided by the Academic Editor). The Author Summary should consist of 2-3 succinct bullet points under each of the following headings:

• Why Was This Study Done? Authors should reflect on what was known about the topic before the research was published and why the research was needed.

• What Did the Researchers Do and Find? Authors should briefly describe the study design that was used and the study’s major findings. Do include the headline numbers from the study, such as the sample size and key findings.

• What Do These Findings Mean? Authors should reflect on the new knowledge generated by the research and the implications for practice, research, policy, or public health. Authors should also consider how the interpretation of the study’s findings may be affected by the study limitations. In the final bullet point of ‘What Do These Findings Mean?’, please describe the main limitations of the study in non-technical language.

INTRODUCTION

1) Please note that references should be preceding punctuation. Also, in some instances, e.g. line 80, the references are still in uppercase format. Please revise throughout the main manuscript.

2) ll.90-94: For clarity, we suggest adding “in mice” after “fibrosarcoma”.

METHODS AND RESULTS

1) Please note that changes in the analysis-- including those made in response to peer review comments-- should be identified as such in the Methods section of the paper, with rationale.

2) Please add the following statement, or similar, to the Methods: "This study is reported as per the Strengthening the Reporting of Observational Studies in Epidemiology (STROBE) guideline (S1 Checklist)."

3) l.247: Please add “IQR” in the parentheses when reporting the interquartile range. Please revise throughout the main manuscript.

4) l.278: “0.75–0.87” – Please replace the hyphen with a comma.

DISCUSSION

1) ll.298-299: “Whether inhibition of factor X also works in other cancer types is unclear.” – We suggest adding here that it is also unclear whether the inhibition of factor X also works in humans given the study was done in mice.

2) ll.306-307: Please mention that these results should also be interpreted with caution due to low event numbers (if correct); perhaps at the end of the sentence on precision?

3) l.347: Please include the reference to Graf et al. Also, please reiterate that the study was a preclinical study in mice.

4) We suggest briefly discussing the transferability from animal models to humans given your study hypothesis is based on data stemming from animal studies.

SUPPLEMENTARY MATERIAL

Thank you for providing the STROBE checklist. Please replace the page numbers with paragraph numbers per section (e.g. "Methods, paragraph 1"), since the page numbers of the final published paper may be different from the page numbers in the current manuscript.

SOCIAL MEDIA

To help us extend the reach of your research, please provide any X (formerly known as Twitter) handle(s) that would be appropriate to tag, including your own, your coauthors’, your institution, funder, or lab. Please respond to this email with any handles you wish to be included when we tweet this paper.

Comments from Reviewers:

Reviewer #2: Thank you to the authors for addressing my previous comments well. I have no further issues to raise.

Reviewer #3: The authors have addressed my previous comments.

[LINK]

General Editorial Requests

---

## [Editor Report · Decision Letter 3]

5 Apr 2024

Dear Dr Bosch, 

On behalf of my colleagues and the Academic Editor, Aadel A Chaudhuri, I am pleased to inform you that we have agreed to publish your manuscript "Direct factor Xa inhibitors and the risk of cancer and cancer mortality: a population-based cohort study" (PMEDICINE-D-23-03433R3) in PLOS Medicine.

I appreciate your thorough responses to the reviewers' and editors' comments throughout the editorial process. We look forward to publishing your manuscript, and editorially there are only a few remaining minor stylistic/presentation points that should be addressed prior to publication. We will carefully check whether the changes have been made. If you have any questions or concerns regarding these final requests, please feel free to contact me at aschaefer@plos.org.

Please see below the minor points that we request you respond to:

1) l.48/l.51: Please change the period to a comma or semicolon after reporting the values of the factor Xa inhibitors cohort.

2) l.52: Please check whether “HR” should be “SHR” here.

3) ll.55-56: Please change to “However, we observed an increase in all-cause mortality in the factor Xa inhibitor cohort.”.

4) Author Summary: Under ‘Why Was This Study Done?’, please change “promote” to “promotes”. Also, under ‘What Did the Researchers Do and Find?’, please remove any values mentioned.

5) In the figure descriptions of Figures 1, 2, and 3, please include the factors that are included in the IPTW. In the figures, please include the numbers at risk and the event numbers for the different cohorts, as well as the subdistribution hazard ratios (you may want to use Figure 1 from https://doi.org/10.1007/s00392-023-02308-y as a template). In the figure description, please include all abbreviations used.

PRESS

Sincerely, 

Alexandra Schaefer, PhD 

Associate Editor 

PLOS Medicine